# Whole-Body Vibration or Aerobic Exercise in Patients with Bronchiectasis? A Randomized Controlled Study

**DOI:** 10.3390/medicina58121790

**Published:** 2022-12-05

**Authors:** Orçin Telli Atalay, Ayşenur Yılmaz, Betül Cengiz Bahtiyar, Göksel Altınışık

**Affiliations:** 1Faculty of Physical Therapy and Rehabilitation, Pamukkale University, 20160 Denizli, Turkey; 2Department of Pulmonology, Denizli State Hospital, 20010 Denizli, Turkey; 3Department of Pulmonology, Faculty of Medicine, Pamukkale University, 20160 Denizli, Turkey

**Keywords:** non-CF bronchiectasis, whole-body vibration, dyspnea, six-minute walk distance, exercise capacity

## Abstract

*Background and Objectives*: The whole-body vibration (WBV) technique is an exercise training method. It has been reported to improve muscle strength, exercise capacity, and the quality of life. However, there is no study on the use of the WBV technique in bronchiectasis. The aim of the present study is to compare the effect of aerobic exercise with whole-body vibration on exercise capacity, respiratory function, dyspnea, and quality of life (QoL) in bronchiectasis patients. *Materials and Methods*: Clinically stable bronchiectasis patients aged 18–74 years participated in this study. A pulmonary function test, 6 minute walk test (6MWT), five times sit-to-stand test (FTSST), Modified Medical Research Council (mMRC) Scale, an, St. Georges Respiratory Questionnaire (SGRQ) were used in the evaluation. In total, 41 patients (WBV group: 20, aerobic group: 21) completed the study. The patients were treated for eight weeks. *Results:* When the two groups were compared after the treatment, there was a significant difference between the mMRC scores in favor of the WBV group (*p* < 0.05). When the results of the WBV group were examined before and after treatment, a significant difference was found between the 5SST and 6MWT (*p* < 0.05). When the aerobic group was compared before and after the treatment, it was observed that there was a significant difference in FVC, FVC%, 5SST, 6MWT, and SGRQ total score, and activity and impact scores, which are the sub-parameters (*p* < 0.05). *Conclusions*: Eight weeks of WBV exercise can lead to significant improvements in patients with bronchiectasis, exercise capacity, and dyspnea. Larger studies are needed to define the optimal intensity and duration of WBV, as well as to investigate its possible long-term effects.

## 1. Introduction

Bronchiectasis is a disease characterized by enlargement of the airways and thickening of the bronchial wall, accompanied by chronic cough and sputum complaints. The recurrence of infective exacerbations in bronchiectasis increases the deterioration of structural integrity, making it difficult for purulent secretions to be removed. Decreased pulmonary functions play a role in bronchiectasis, which is an important part of morbidity and mortality, and is directly proportional to the increase in secretions. In addition, functional capacity and QoL decrease in patients with bronchiectasis [1].

The effectiveness of pulmonary rehabilitation programs in reducing the burden of the disease has been reported in previous studies [2,3]. Exercises that can minimize dyspnea (aerobic and resistance training) can increase dyspnea when the patient is forced, affecting the treatment program. The patient may not want to exercise due to fear of shortness of breath. Patients who are intolerant to exercise cannot, therefore, fully benefit from a traditional pulmonary rehabilitation program [3,4]. WBV has been used as a therapeutic approach to improve muscle strength as a more tolerable training alternative for patients with chronic obstructive pulmonary disease (COPD) [5].

Bronchiectasis patients may have a lower tolerance to resistive exercise; thus, it appears to be a reasonable alternative to traditional training programs. WBV is a mode of physical activity. The person stands on the vibration platform, which mainly accelerates in the vertical direction. Physiologically, similar responses are seen in aerobic and resistance training. It is thought that WBV increases the effects of spinal reflex mechanisms with the principle of muscle activation and contributes to improving muscle strength [6,7].

Studies have reported that leg muscle strength, general muscle strength, exercise capacity [5,8,9], and oxygen consumption [10] increase after WBV. Pulmonary function tests (PFT) are uncertain and require further investigation [5,11]. The effects of WBV on QoL have also been investigated; several authors reported that WBV exercise can improve the QoL of individuals with COPD [12,13]. WBV has been reported to be safe, convenient, and feasible under controlled conditions [14,15,16]; no side effects have been observed in previous studies [14,17,18,19]. However, no studies on this subject in patients with bronchiectasis have been conducted to date. 

The aim of this study was to evaluate the effect of WBV on exercise capacity, respiratory functions, dyspnea, and QoL in patients with bronchiectasis, and to compare its effectiveness with aerobic exercise.

## 2. Methods

### 2.1. Trial Design

This randomized study was performed to evaluate the eight weeks of WBV exercise in Bronchiectasis patients. The individuals were randomly allocated to a WBV group or an aerobic group. The research was carried out between February 2017 and July 2022. With the outbreak of the COVID-19 pandemic process in Turkey in March 2020, outpatient physiotherapy and rehabilitation services were completely stopped for about 3 months. Subsequently, rehabilitation services were interrupted from time to time for about 1 year. After the effect of the pandemic had eased, patients with respiratory disease in particular, as well as those with bronchiectasis, refused to participate in rehabilitation programs at the hospital for a significant period of time. Although some of the data collection was finished before the pandemic, reaching the necessary participants and completing the treatment and evaluation processes meant that the study had to be extended until 2022.

This study protocol was approved by the ethics committee of P Date: Pamukkale University (No:60116787-020/13943- 21 February 2017). The research protocol has been compatible with the Declaration of Helsinki and informed consent has been obtained from all participants.

### 2.2. Inclusion Criteria 

Patients who were aged 18 years and over, clinically stable (not having any exacerbation in less than 4 weeks), and with a diagnosis of non-CF bronchiectasis were included in the study.

### 2.3. Exclusion Criteria

The exclusion criteria for the study were: those with a history of pneumothorax, myocardial infarction, and surgery; those who had cor pulmonale and/or heart failure, hemoptysis, respiratory distress requiring hospitalization, spinal cord injury, unstable intervertebral discs or rib fracture; those who exhibited an infective exacerbation during the physiotherapy, who were suffering from a comorbid disease which might be a contraindication for exercise, chest physiotherapy (advanced osteoporosis, vertigo, neurologic diseases, etc.), and orthopedic injuries.

### 2.4. Randomization

Groups were determined using the closed envelope method. The patients did not know how many groups there were, or which group they were in. The same physiotherapist supervised the physiotherapy programs of the two groups. Measurements were conducted by another physiotherapist blinded to intervention allocation at both the beginning and end of the study.

### 2.5. Participants

This study was completed with 41 patients, who met the inclusion criteria, and voluntarily agreed to participate in the study; 20 were assigned to the vibration group, and 21 were assigned to the aerobic exercise group. Patients who were diagnosed with bronchiectasis at Pamukkale University Chest Disease Outpatient Clinic and fulfilled the inclusion criteria were included in this study. 

## 3. Measurements

### 3.1. Evaluation

The sociodemographic characteristics of the participants who met the inclusion criteria were determined through mutual interviews. Sputum analysis was performed by our doctors in the evaluation of all the patients. Additionally, it was used in Bronchiectasis Severity Index (BSI) scoring. The BSI was used for identifying the severity of bronchiectasis as mild, moderate, and severe. After the descriptive information of the participants was recorded, the lung function of all participants was determined by the PFT; dyspnea level was assessed with the mMRC Scale; exercise capacity was measured with the 6MWT and FTSST, and QoL was recorded through the SGRQ. Evaluations were performed twice: before and after the treatment.

#### 3.1.1. Pulmonary Function

PFT was performed with the COSMED Pony Fx portable device according to the ATS/ERS guidelines. Forced expiratory volume in the first second (FEV1), forced vital capacity (FVC), FEV1/FVC ratio, and peak expiratory flow (PEF) values were recorded [20].

#### 3.1.2. Dyspnea

Our patients’ before and after treatment mMRC scores were requested by reading the scale options to the patient and choosing the most appropriate degree to describe the patient’s respiratory distress [21,22].

#### 3.1.3. Exercise Capacity

The FTSST is used to evaluate exercise capacity. Studies have reported that the results are correlated with the 6MWT. The patients were asked to sit with their arms crossed over their shoulders and their back leaning on the chair. They were asked to stand up and sit down quickly five times from a standard chair with a height of 43 cm. Their best times were recorded [23].

The 6MWT was performed according to the American Throracic Society (ATS) statement. It is used to measure functional capacity in people with lung disease and to evaluate the response to pulmonary rehabilitation. The distance the patient walked in 6 min was recorded in meters [24].

#### 3.1.4. Quality of Life

Consisting of 50 items and 76 different scored answers, the SGRQ is a questionnaire that evaluates health-related QoL with three subscales of symptom, activity, and impact. Scores ranging from “0 to 100” are calculated for each subscale and a total score is obtained. A score of ‘0’ indicates that there is no impairment in the quality of life, and an increase in the score indicates that the QoL is affected [25,26].

## 4. Intervention

The subjects were divided into two groups by randomization using the closed envelope method. Respiratory physiotherapy, posture exercises, and WBV for 10–15 min (Vibration group) were assigned to the subjects in the first group, and respiratory physiotherapy and posture exercises were assigned to the subjects in the second group. Aerobic exercise was performed on a treadmill (aerobic exercise group) for 30 min. The treatment was applied for 8 weeks, 3 days a week for both groups by the same supervisor physiotherapist. The duration of the program was planned as 6 weeks at first, but because the results obtained from the pilot study suggested that this period might be short, and recent studies reported the optimum duration for pulmonary rehabilitation programs to be 8–12 weeks, the duration of the treatment program was extended to 8 weeks. Evaluations were conducted by another physiotherapist on the first and last day of treatment. The treatments were performed face to face. Additionally, the exercises were conducted individually. The exercises were performed between 9.00 and 18.00, depending on the patient’s availability. Additionally, the sessions lasted approximately 1 h.

### 4.1. Aerobic Exercise 

The exercise intensity on the treadmill was gradually increased and adjusted to reach 60–85% of the maximum heart rate. Each session was divided into 5 min of warm-up, 20 min of exercise, and 5 min of cool-down. Heart rate and O_2_ saturation were measured at the beginning, middle, and end of the treatment [27].

### 4.2. Vibration Training

For WBV, low-frequency and low-amplitude mechanical vibration have been reported to be an effective and safe method of improving muscle strength. WBV exercises, defined as exercises applied to the body in contact with a vibrating platform, can benefit the knee joint by stimulating reflex muscle contractions and synchronizing motor unit activation [16,28,29]. The advantage of WBV is that it reduces rehabilitation times compared with other traditional treatment programs. 

WBV was performed with a vertical vibration platform (Power Plate Pro5^®^, Northbrook, IL, USA). With a vibrating platform, the individual moves the right and left legs upwards with a frequency of 25–50 Hz and a range of 6 mm [29,30]. The standard calibration was used as set by the manufacturer. There was no electronic recording device used in every session. The patients stood with their knees bent at approximately 20 degrees during the WBV. Participants were instructed not to hold on to the bars during the WBV. The treatment protocol for the first 4 weeks was a 5–10 min cycling warm-up, then 30 Hz vibration for 30-s stand in squatting position, then 30-s rest for 10 min; in the last 4 weeks, 30 Hz vibration was applied in squatting position for 60 s, then 30-s rest for 15 min. In the treatment, the vibration frequency was kept constant and the time was increased. Leg fatigue and shortness of breath were questioned with the Borg scale. Leg fatigue, shortness of breath, heart rate, and oxygen saturation were measured before and after each treatment [31].

In addition, respiratory physiotherapy and posture exercises were performed in both patient groups. Stretching exercises for the pectoral muscles, strengthening the scapula, and thoracic extensors were directed as posture exercises. 

## 5. Sample Calculation

It was observed that the effect size obtained in the reference study was at a strong level (d = 1.13; based on the 6MWT distance) [31]. Based on the results of the reference study, assuming that we could obtain a lower effect size (d = 1), as a result of the power analysis, it was calculated that 95% power could be obtained at the 95% confidence level when at least 46 people (23 for each group) were included in the study. 

## 6. Analysis

The data were analyzed with the SPSS package program. Continuous variables are presented as the mean ± standard deviation, and categorical variables are provided as numbers and percentages. Initially, we performed the Shapiro–Wilk normality test for all variables. It does not provide parametric test assumptions; therefore, the Mann–Whitney U test was applied for comparisons of independent group differences. The Wilcoxon test was used to compare dependent groups. To evaluate the minimum clinically important difference (MCID) for walking distance, an increase of 35 m was considered at the end of training the groups [32]. Subsequently, patients were categorized as ‘reached the MCID’ and ‘not reached the MCID’, and comparisons to the frequency of occurrence between the WBW and aerobic groups were conducted using Fisher’s exact test. Data analysis was performed using SPSS software (SPSS Inc., Chicago, IL, USA) version 20.0, and a significance level of 5% (*p* < 0.05) was adopted for all tests.

## 7. Results

Fifty-one patients were recruited, and 44 patients fulfilled the eligibility criteria. Four volunteers did not meet the inclusion criteria (two patients have neurologic diseases, one patient has a history of myocardial infarction, and one had heart failure), and three volunteers did not want to participate in the study. Forty-four patients were randomized. As a result of randomization, 22 patients were in the vibration group, and 22 patients were in the aerobic group. After starting the treatment, two patients in the vibration group could not continue because their work hours were not suitable. Exacerbation was observed in one patient in the aerobic exercise group. The flow chart of our study is depicted in Figure 1.

Underlying etiologies of non-CF bronchiectasis were autoimmune diseases in 5 (12.2%), idiopathic in 6 (14.6%), infection in 1 (2.5%), tuberculosis in 1 (2.5%), and childhood infections in 28 (68.2%) subjects (19 (67.9%) pneumonia, 6 (21.4%) tuberculosis, and 3 (10.7%) whooping cough). Of the subjects, 28 (68.3%) had no smoking history and 13 (31.7%) patients were ex-smokers. According to high-resolution computed tomography reports, 19 (46.4%) of the subjects had one affected lobe, 3 (7.3%) had two affected lobes, 12 (29.2%) had three affected lobes, 1 (2.5%) had four affected lobes, and all lobes were affected in 6 (14,6). There were 26 (63.4%) mild, 11 (26.8%) moderate, and 4 (9.8%) severe patients according to the Bronchiectasis Severity Index. Of the patients, 28 (68.2%) were using bronchodilators, and 18 (43.9%) were using inhaled corticosteroids. Pulmonary function testing revealed that patients had moderate airflow obstruction (Table 1). There were 35 patients (85.4%) with FEV1 results below 80% of predicted. When sputum cultures were examined, 10 patients (50%) in the WBV group had pseudomonas infection and 14 patients (66.7%) in the aerobic group had pseudomonas infection.

The mean age of the WBV group was 55.66 ± 17.10 years and the mean age of the control group was 41.66 ± 15.41 years (*p* > 0.05). The mean body mass index was 26.71 ± 3.29 kg/m^2^ in the WBV group and 22.96 ± 2.91 kg/m^2^ in the aerobic exercise group (*p* > 0.05). Respiratory parameters, 6MWT results, FTSST, and QoL results were similar in the two groups before the treatment (*p* > 0.05). Demographic clinical characteristics and baseline test results of the participants are shown in Table 1. 

The descriptive information of the participants was similar between the groups (*p* > 0.05). There was only a significant difference between the two groups in terms of weight (*p* < 0.05). In the WBV group, 9 of the participants were male and 11 were female; in the aerobic exercise group, 8 of the participants were male and 13 were female. The patients had no smoking or alcohol use. There was no significant difference between disease duration and mMRC score (*p* > 0.05). 

When the results of the WBV group were examined before and after the treatment, there was a significant difference in the FTSST and 6MWT. However, there were significant differences in FVC, FVC%, FTSST, and 6MWT in the aerobic group, total score, and in activity and effect scores, which are sub-parameters of the quality of life (*p* < 0.05). After the treatment, a significant difference was determined only in the total score of QoL and activity, which is one of its sub-parameters (*p* < 0.05; Table 2). When the two groups were compared after the treatment, there was a significant difference between the mMRC scores (*p* < 0.05). 

When the two groups were compared, no differences were observed between the two groups in the 6MWT distance change (WBV group 44.91 ± 35.88 m, aerobic group 40.97 ± 25.34 m, *p* = 0.426). The MCID for the distance walked on 6MWT was obtained in 42.1% and 77.8% of patients in the aerobic and WBV groups, respectively (χ^2^ = 4.880, *p* = 0.027).

## 8. Discussion

This randomized, controlled study investigated the effects of WBV in bronchiectasis and also compared the effects of WBV and aerobic exercise on exercise capacity, respiratory functions, dyspnea, and QoL in patients with bronchiectasis. It was shown that there We added the section number. Please confirm.were significant increases in terms of exercise capacity for both exercise types after 8 weeks of intervention. The dyspnea was found to be significantly reduced only in the WBV group. However, the improvement in quality of life was significantly higher in the aerobic exercise group.

Whole-body vibration is known to be a dyspnea-free exercise [13]. According to previous studies on the effects of WBV on COPD, it was emphasized that due to lung emphysema and chronic bronchitis, COPD patients suffer from severe dyspnea, especially during exercise. WBV in COPD patients has been a safe exercise at all frequencies and types of squats, without causing dyspnea and fatigue [12,13,33]. This may also be true for bronchiectasis patients. In this study, t was observed, but not measured, that dyspnea was induced in the aerobic exercise group especially during the first few treatment sessions, due to the increased oxygen demand. However, in the WBV group, the perceived difficulty level was lower, and the subjects in this group adapted to exercise quickly. Some previous studies have reported that WBV exercise cannot only enhance physical status, but also decrease fatigue in various populations [34,35,36]; we believe that this may also affect the perception of dyspnea. The “tonic vibration reflex” caused by mechanical vibration induces a higher rate of motor unit recruitment in skeletal muscles by stimulating muscle spindles and Golgi cells. This provokes muscle contractions and leads to an increase in muscle strength, especially in the lower extremities, producing less fatigue when compared with voluntary muscle contraction. In patients with stable COPD, it has been shown that WBV does not alter oxygen saturation [37]. The oxygen saturation was measured at the beginning and end of the exercise in both groups to monitor the subjects during the exercise but was not analyzed as an outcome measure in this study. However, we believe that another reason for lower dyspnea perceptions may be lower alterations in oxygen saturation in the WBV group compared with the aerobic exercise group.

The increase in the 6MWT results in our bronchiectasis patients who received an 8 week WBV program was consistent with previous findings. When the results of the 6MWT after WBV were examined in COPD patients, it was observed that walking distance increased [30]. In terms of the 6MWT, our patients achieved the MCID for walking distance, by Puhan et al. These researchers reported that at least a 35 m difference should exist before and after treatment in the 6MWT for the results to be clinically significant [32]. In our study, an average increase of 44.91 m was found within walking distance in the WBV group in the 6MWT. Additionally, a significant increase of 40.97 m was found in the aerobic group. The WBV group improved the walking distance more than the aerobic group, but it was not significantly different. 

Although it is not statistically meaningful, the average ages of subjects in the groups were different. The older group was those performing WBV, but they tolerated the exercise well. The WBV is a commonly used exercise modality for elderly participants for several reasons, such as significant improvements in dynamic and static balance, postural control, muscle strength, bone density, physical fitness, and functionality [5,31,32]. If the average age of the aerobic exercise group was older, the improvements in functional capacity might be thought to be lower as a result of age. However, the baseline functional capacity is more important than age in terms of responses to exercise [38]. The pre-exercise capacities were similar in the two groups in our study. The lack of difference in 6MWT results between the aerobic exercise program and WBV and the similarly significant increases in both groups compared with pre-exercise showed that WBV may be a good alternative treatment option to aerobic exercise in patients with bronchiectasis in terms of increasing exercise capacity without inducing dyspnea.

A reduction in time to complete the FTSST has been reported after three weeks of resistance training combined with WBV in COPD patients [37]. In our study, a decrease in the FTSST duration was observed in both aerobic exercise and WBV groups. It has been shown in many studies that aerobic exercise improved FTSST results in COPD [37]. In a recent study, it was reported that following PR, there were significant improvements in FTSST in bronchiectasis but as the change in FTSST did not correlate with changes in other outcome measures and it was suggested to just provide additional information to traditional PR outcome measures [23]. We, therefore, used both 6MWT and FTSST for measuring exercise capacity. There is not any other study investigating the effects of WBV on exercise capacity with FTSST. 

In our study, no significant changes were found in pulmonary function in the WBV group. To our knowledge, there has not been any study about the effects of WBV on pulmonary function in bronchiectasis; however, a recent systematic review examining the effects of WBV on pulmonary function in COPD reported that no great benefits on pulmonary function were found, similarly to our results [11]. 

On the other hand, there was only a significant change in FVC, but not in FEV1, in the aerobic exercise group. In a recent meta-analysis performed by Ora et al., the effects of pulmonary rehabilitation, including aerobic exercises in non-cystic fibrosis bronchiectasis, revealed that the FEV1 assessment after PR, did not show any significant increase between the active and control group and PR improved exercise tolerance in non-cystic fibrosis bronchiectasis patients, but it has a modest impact on respiratory function [39]. One study showed that limb cycle exercise at 75% of peak O_2_ resulted in a significant improvement in FVC, but not in any other spirometric parameters [40]. This increase was explained as a result of improvements in inspiratory capacity and respiratory muscle strength. We also think that the improvement in FVC, but not in FEV1, after aerobic exercise training could be a result of an improvement in inspiratory capacity and an increase in respiratory muscle strength, because there might be an increase in the maximal shortening of inspiratory muscles as an effect of training which might have led to an increase in inspiratory volume. However, further studies with a larger population focusing on spirometric parameters in bronchiectasis are needed to explain this.

When the QoL was compared, a significant difference was observed between the two groups in favor of the aerobic group. In addition, a significant difference was found in the aerobic group before and after treatment in the activity and effect sections of SGRQ and its sub-dimensions. Additionally, improvements in QoL were achieved in the WBV group when comparing before and after treatment, although it was not significant. The QoL was assessed with SGRQ. One disadvantage of questionnaires is that they are based on patients’ statements which can be related to the patient’s level of satisfaction and what the patient expects from the treatment. This can be a reason for the difference in QoL between the two groups. Another explanation may be that, although the mean age of subjects was not significantly different between the two groups, the subjects in the WBV group were older than the subjects in the aerobic exercise group and as reported in a previous study, the quality of life of elderly patients is affected more, recovery is more difficult and increasing age is among the factors that negatively affect the quality of life [41]. In a 12-week study, improvements in all aspects of QoL and increased walking distances were reported at the end of the WBV program for patients with COPD [36]. Long-term studies may be suggested to show the effect of WBV on the quality of life. We assume that evaluation 3 months after the treatment or later may affect the results. It may be related to the duration and severity of WBV. 

To the best of our knowledge, this study is the first to investigate the effects of WBV on bronchiectasis. In addition, the effects of WBV were compared with aerobic exercise which is a commonly used exercise type in PR. The patients with bronchiectasis were very satisfied with the WBV treatment because of the shorter training time—less feeling of fatigue and shortness of breath. This intensity and duration were applied safely without any adverse effects. However, studies on different intensities and durations are needed. 

We think that the vibration applied at a certain frequency with WBV will provide transitional vibration and/or shaking on the thoracic region, similar to chest physiotherapy techniques for sputum movement and clearance, there are not any studies about this subject. As an observational result, with the verbal feedback we received from patients in the WBV group, they stated that sputum excretion increased at the beginning and they produced sputum easily; then, they stated that sputum production was still comfortable, and the amount of sputum decreased. Patients in the aerobic exercise group stated that there was not much change in sputum. 

### Limitations of the Study

The effect of WBV on cardiopulmonary responses such as heart rate and oxygen saturation, as well as peripheral muscle strength, respiratory muscle strength, fatigue, and sputum amount was not examined in this study. A study with a placebo group would have provided a better view of the effect of WBV. Further studies with larger sample sizes are suggested to focus on these aspects as well as QoL in patients with bronchiectasis

## 9. Conclusions

The results of this study showed that WBV can be as effective as aerobic exercise on dyspnea and functional capacity, and is suggested as an easy, shorter, and enjoyable choice of exercise modality in the rehabilitation of bronchiectasis. Larger studies are needed to define the optimal intensity and duration of WBV, as well as to investigate its possible long-term effects.

## Figures and Tables

**Figure 1 medicina-58-01790-f001:**
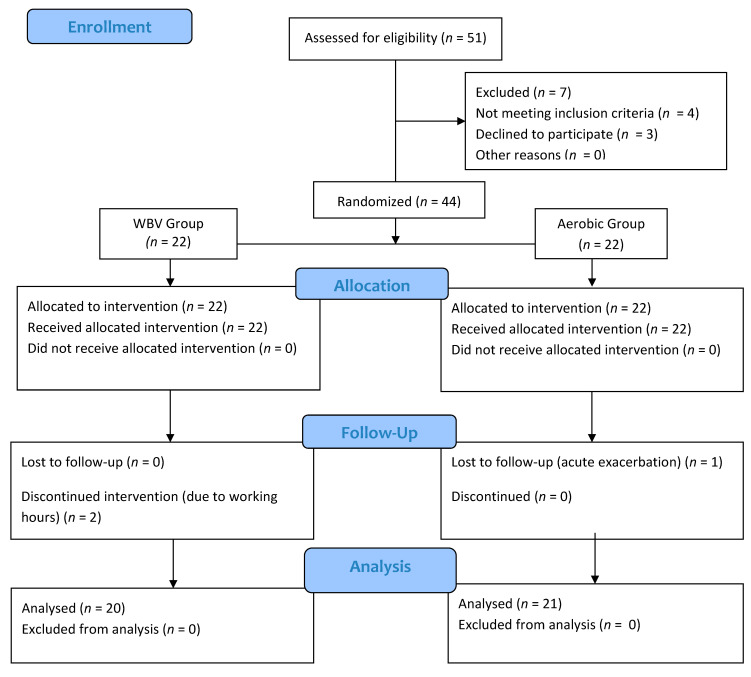
CONSORT diagram for study flow. Flow diagram illustrating the number of participants in each group.

**Table 1 medicina-58-01790-t001:** Comparison of two groups in terms of baseline demographic, clinic and functional characteristics.

Variables	WBV Group (*n* = 20) X ± SD	Aerobic Exercise Group (*n* = 21) X ± SD	Z	*p* Value
Age (years)	55.66 ± 17.10	41,66 ± 15.41	−1.949	0.051
Height (m)	163 ± 9.58	164 ± 7.6	−0.581	0.561
Weight (kg)	71.1 ± 10.64	61.55 ± 5.98	−1.990	0.047 *
BMI (kg/m^2^)	26.71 ± 3.29	22.96 ± 2.91	−2.033	0.400
Smoking history (pack years)	4.22 ± 4.17	3.11 ± 4.91	−0.584	0.559
Number of exacerbations per year (n)	1.55 ± 0.88	0.77 ± 0.66	−1.927	0.054
Disease severity ^a^				
Mild bronchiectasis (0–4)	3 ± 1.14	2.75 ± 1.25	−0.139	0.889
Moderate bronchiectasis (5–8)	6 ± 1.15	6.66 ± 1.54	−0.727	0.467
Severe bronchiectasis (9+)	9.66 ± 1.15	9.5 ± 0.7	−1.291	0.197
Disease Duration (years)	16.44 ± 13.67	25.33 ± 15.16	−1.552	0.121
mMRC	2.55 ± 1.01	2.44 ± 2.06	−0.142	0.887
FEV1 (L)	1.59 ± 0.44	1.93 ± 0.46	−1.372	0.170
FEV1 (%)	58 ± 14.23	59.44 ± 12.78	−0.310	0.757
FVC (L)	2.25 ± 0.6	2.53 ± 0.53	−0.884	0.377
FVC (%)	66.88 ± 14.17	69.22 ± 11.36	−0.623	0.533
FEV1/FVC (%)	70.33 ± 8.21	71.88 ± 8.1	−0.399	0.690
PEF (L/sn)	3.89 ± 1.83	4.46 ± 1.46	−0.833	0.377
PEF (%)	50.76 ± 16.68	57.55 ± 12.88	−0.708	0.479
6 MWT (m)	433 ± 67.61	432.88 ± 96.22	−0.132	0.895
FTSST (seconds)	11.75 ± 3.75	10.85 ± 2.6	−0.574	0.566
Saint georges respiratory questionnaire				
Total	49.88 ± 13.18	54.21 ± 18.93	−0.662	0.508
Activity	58.39 ± 26.64	66.72 ± 17	−0.487	0.626
Symptom	53.06 ± 18.07	61.1 ± 22.38	−0.751	0.453
Impact	44.02 ± 13.72	44.91 ± 24.02	−0.397	0.691

BMI—body mass index; mMRC—modified Medical Research Council; FEV1—forced expiratory volume in first second; FVC—forced vital capacity; PEF—Peak Expiratory Flow; FTSST—Five-times-sit-to-stand test; 6MWT—6-Minute Walking Test; ^a^ Disease severity based on Bronchiectasis Severity Index (BSI); * *p* < 0.05; Man—Whitney u test.

**Table 2 medicina-58-01790-t002:** The effect of treatment on functional exercise capacity, dyspnea, respiratory functions, and disease severity.

Variables	WBV Group (*n* = 20)				Aerobic Exercise Group (*n* = 21)				Differences between Groups	
	Baseline X ± SD	Post Treatment X ± SD	Z	*p* ^a^	Baseline X ± SD	Post Treatment X ± SD	z	*p* ^a^	z	*p* ^b^
FEV1 (L)	1.59 ± 0.44	1.53 ± 0.45	−0.178	0.858	1.93 ± 0.46	1.91 ± 0.41	−0.421	0.674	−1.680	0.093
FEV1 (%)	58 ± 14.23	56.32 ± 16.24	−0.060	0.952	59.44 ± 12.78	61.77 ± 11.06	−1.562	0.118	−0.532	0.595
FVC (L)	2.25 ± 0.6	2.26 ± 0.6	−0.420	0.674	2.53 ± 0.53	2.64 ± 0.58	−2.207	0.027 *	−1.546	0.122
FVC (%)	66.88 ± 14.17	66.88 ± 14.15	0.000	1.000	69.22 ± 11.36	73.68 ± 8.57	−2.207	0.027 *	−1.683	0.092
FEV1/FVC (%)	70.33 ± 8.21	68.44 ± 9.90	−0.178	0.859	71.88 ± 8.1	72.98 ± 13.67	−0.315	0.752	−0.972	0.331
PEF (L/sn)	3.89 ± 1.83	4.07 ± 2.06	−0.980	0.327	4.46 ± 1.46	4.84 ± 1.73	−0.170	0.865	−0.751	0.453
PEF (%)	50.76 ± 16.68	51.81 ± 15.37	−0.416	0.677	57.55 ± 12.88	54.21 ± 18.93	−1.272	0.203	−1.679	0.093
6 MWT (m)	433 ± 67.61	477.91 ± 41.21	−2.310	0.021 *	432.88 ± 96.22	473.85 ± 99.58	−2.668	0.008 *	−0.132	0.895
FTSST (seconds)	11.75 ± 3.75	10.18 ± 3.23	−2.429	0.015 *	10.85 ± 2.6	9.19 ± 2.42	−2.547	0.011 *	−0.927	0.354
mMRC	2.55 ± 1.01	0.78 ± 0.8	−2.724	0.006	2.44 ± 2.06	3 ± 2.11	−1.732	0.083	−2.703	0.007
SGRQ										
Total	49.88 ± 13.18	43.55 ± 14.61	−0.700	0.484	54.21 ± 18.93	27.65 ± 9.82	−2.666	0.008 *	−2.075	0.038 *
Activity	58.39 ± 26.64	54.20 ± 16.47	−0.420	0.674	66.72 ± 17	32.22 ± 14.23	−2.666	0.008 *	−2.433	0.015 *
Symptom	53.06 ± 18.07	48.33 ± 17.55	−0.980	0.327	61.1 ± 22.38	44.32 ± 17.48	−1.400	0.161	−0.662	0.508
Impact	44.02 ± 13.72	35.98 ± 14.61	−1.120	0.263	44.91 ± 24.02	19.82 ± 13.71	−2.240	0.025 *	−1.898	0.058

FEV1—forced expiratory volume in first second; FVC—forced vital capacity; PEF—Peak Expiratory Flow; FTSST—Five-times-sit-to-stand test; 6MWT—6-Minute Walking Test; SGRQ—Saint George’s respiratory questionnaire; ^a^ Wilcoxon rank test; ^b^ Mann—Whitney U-test; significance level; * *p* < 0.05.

## Data Availability

Not applicable.

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
