# Peer review of "Whole-Body Vibration or Aerobic Exercise in Patients with Bronchiectasis? A Randomized Controlled Study"

_medicina, 2022, doi:10.3390/medicina58121790_

Round 1
Reviewer 1 Report
Dear editor,
In this study, the writers compared prospectively, 2 different interventions for treating adult patients with non CF bronchiectasis.
They found that both aerobic exercise and whole body vibrations increased 6MWT, and improved 5SST, with no significant difference between the groups.
They conclude that WBV can be an alternative method to aerobic exercise for bronchiectasis patients
The manuscript needs to be re written. The structure needs to be improved, and the English needs to be fixed.
Points:
The study is stated as "single blind", and it is commented that the patients did not know in what group they were (line 100), however since they were doing only one intervention, it is not possible that they were not aware of what they were doing and therefore they were not blinded, regardless of the fact that they were not aware of other interventions for other patients.
The research was carried out between 2017 and 2022 – please comment on if and how did the covid pandemic affect this study.
In the Methods – for the inclusion criteria, only patients that did not experience an exacerbation were included. It is not clear if those with very recent exacerbations were excluded, and if so, how many weeks from last exacerbation needed.
Those with a recent MI were excluded – how recent? Any time in life or in the past months?
Those with exacerbation during the intervention are listed as excluded, however they cannot be excluded since they were already enrolled for intervention.
In lines 102 till 133 the methodology of the different tests is described in detail, I would suggest referencing the tests and avoiding long descriptions. (it is enough to say that PFTs were conducted according to AST ERS requirements and so on… )
In line 139 – the treatment was applied for 8 weeks… in the poster manuscript presented in 2019 by the writers, the patients were followed for 6 weeks, please explain if protocol was changed along the way and if so please rationalize. (DOI: 10.1183/13993003.congress-2019.PA1176)
On the consort diagram it is not clear what side of the diagram belongs to what intervention
Sample calculation – needs to be reworded.
For the results:
Please explain what childhood illness refers to. (post infectious? PCD? Retained foreign body?...
It seems like many of the patients had only one lobe involved, would you expect differences between those with one lobe involved compared with those that have a more systemic disease?
In table 1 p value for age is 0.510, but in line 217 it is less than 0.05, please double check.
Line 231 – there was a significant change in FVC only in the aerobic group. Please rationalize why this was not seen in the WBV group and why this effect was seen in FVC but not in FEV1. May this be a type I error?
How would you rationalize the significant decrease in mMRC score only in the WBV group?
Did you have sputum collected for the patients, any difference in microbiology? (for instance pseudomonas infection)
How do you think the fact that the exercise group were younger and had less exacerbations per baseline influenced your results?
Were patients asked to refrain from aerobic exercise at time of the study? Is it possible that some of the patients were also exercising in other times outside of the study?
The discussion section needs to be re written, and English fixed.
For the conclusion –
Since there is profound evidence that aerobic exercise is beneficial in bronchiectasis, and by far less evidence for WBV, would you not be concerned that patients will prefer the "easy" intervention over the intervention that has clear advantages, not only for bronchiectasis but also for other cardiovascular aspects?
Author Response
02/11/2022
Manuscript ID:
Title: Whole-body vibration or aerobic exercise in patients with
bronchiectasis?: A Single blind randomized controlled study
Dear Editor,
First of all, we would like to thank you for giving a chance to revision of the our manuscript. Taking the suggestions and the constructive criticism of the valuable referees into account, we have tried to do our best in the revision of article. We would also like to thank the referees for the guidance and the contributions to article.
Kind regards,
Reviewers'comments:
Reviewer #1:
Dear Reviewer,
First of all thank you for your valuable contribution for improving the manuscript. We tried to revise the mansucript according to your suggestions and for some of them we tried to write the possible explanations as listed below.
Sincerely.
Question 1: The study is stated as "single blind", and it is commented that the patients did not know in what group they were (line 100), however since they were doing only one intervention, it is not possible that they were not aware of what they were doing and therefore they were not blinded, regardless of the fact that they were not aware of other interventions for other patients.
Correction 1: We revised the explanation of blinding in the related line (100) “The patients did not know how many groups there were and which group they were in. Groups were determined by the closed envelope method. Physiotherapist treating and randomizing were different” as “ Groups were determined by closed envelope method. The patients did not know how many groups there were and which group they were in. The same physiotherapist supervised the physiotherapy programs of the two groups. Measurements were made by another physiotherapist blinded to intervention allocation both at the begining and end of the study. (This was also added consort diagram)
Question 2: The research was carried out between 2017 and 2022 – please comment on if and how did the covid pandemic affect this study.
Correction 2:The explanation below is added in the related section
With the start of the Covid pandemic process in Turkey in March 2020, outpatient physiotherapy and rehabilitation services were completly stopped for about 3 months. Afterwards, rehabilitation services were interrupted from time to time for about 1 year. After the effect of the pandemic eased, especially patients with respiratory diesease as well as bronchiectasis refused to participate in the rehabilitation programs at the hospital for a while . Although some of the data collection was finished until the pandemic process, reaching the necessary participant and completing the treatment and evaluation processes in order to complete the study caused the study to be extended until 2022.
Question 3: In the Methods – for the inclusion criteria, only patients that did not experience an exacerbation were included. It is not clear if those with very recent exacerbations were excluded, and if so, how many weeks from last exacerbation needed.
Correction 3: Inclusion crtieria is revised as;
“Clinically stable (not having any exacerbation in less than 4 weeks)”
The pulmonologists in our study preferred not to refer those patients as they thought that it would be unsafe. Also in previous studies it was reported like this.
Question 4: Those with a recent MI were excluded – how recent? Any time in life or in the past months?
Correction 4: Inclusion crtieria is revised as;
Those with a history of myocardial infarction
(there was not any patient with MI history in our study)
Question 5: Those with exacerbation during the intervention are listed as excluded, however they cannot be excluded since they were already enrolled for intervention.
Correction 5:The subject who had an exacerbation while the study was continuing (the program started but the subject did not complete because of exacerbation) was excluded.It was shown in the follow up in consort.
Question 6: In lines 102 till 133 the methodology of the different tests is described in detail, I would suggest referencing the tests and avoiding long descriptions. (it is enough to say that PFTs were conducted according to AST ERS requirements and so on… )
Correction 6:The description of well known tests were shortened in accordance with your suggestion.
The reference is renewed also.
Question 7: In line 139 – the treatment was applied for 8 weeks… in the poster manuscript presented in 2019 by the writers, the patients were followed for 6 weeks, please explain if protocol was changed along the way and if so please rationalize (1).
Correction 7:The duration of the program was planned as 6 weeks at first but as the data obtained from the pilot study suggested that this period might be short (1), and the recent previous studies reported the optimum duration for pulmonary rehabilitation programs as 8-12 weeks, the duration of the treatment program was extended to 8 weeks.
1-Atalay, O. T., Yilmaz, A., Altınışık, G., Cengiz, B., Taşkın, H., Yalman, A., & Kızmaz, E. (2019). Comparison of the effect of aerobic exercise with whole body vibration in patients with bronchiectasis: Single Blind Randomized Controlled Study. DOI: 10.1183/13993003.congress-2019.PA1176.
Question 8: On the consort diagram it is not clear what side of the diagram belongs to what intervention
Correction 8:The consort diagram is revised
Question 9: Sample calculation – needs to be reworded.
Correction 9: Sample size is recalculated and written as below;
It was observed that the effect size obtained in the reference study was at a strong level (d=1.13; Based on 6MWT) (1). Based on the results of the reference study, assuming that we could obtain a lower effect size (d=1), as a result of the power analysis, it was calculated that 95% power could be obtained at the 95% confidence level when at at least 46 people (23 for each group) were included in the study.
The related reference has been changed
1.Greulich, T., Nell, C., Koepke, J., Fechtel, J., Franke, M., Schmeck, B., ... & Koczulla, A. R. (2014). Benefits of whole body vibration training in patients hospitalised for COPD exacerbations-a randomized clinical trial. BMC pulmonary medicine, 14(1), 1-9.
Question 10 Please explain what childhood illness refers to. (post infectious? PCD? Retained foreign body?...
Correction 10: Childhood illness refers to childhood infections so it has beeen changed as childhhod infections and the numbers of specific infection are given:
(Underlying etiologies of non-CF bronchiectasis were autoimmune diseases in 5, idi-opathic in 6, infection in 1, tuberculosis in 1 and childhood infection in 28 subjects ( 19 pneumonia, 6 tuberculosis, 3 whooping cough)
Question 11: It seems like many of the patients had only one lobe involved, would you expect differences between those with one lobe involved compared with those that have a more systemic disease?
Correction 11: Altgough the spirometric changes is minimally affected in when one lobe involved,the airway obstruction is more evident in those patients. So in these patients exercise capacity may be lower. But although the negative effects of bronchiectasis on exercise capacity and daily activities are known, there is limited information about the physical function evaluation of the effect of disease severity and spread of the disease on exercise capacity. So it is hard to explain the differences in terms of affected lobes.
Question 12: In table 1 p value for age is 0.510, but in line 217 it is less than 0.05, please double check.
Correction 12: The p value has been changed as <0.05
Question 13: Line 231 – there was a significant change in FVC only in the aerobic group. Please rationalize why this was not seen in the WBV group and why this effect was seen in FVC but not in FEV1. May this be a type I error?
Correction 13: In our study no sginificant chages were found in pulmonary function in WBV group. There is not any study about the effects of WBV on pulmonary funtion in bronchiectasis but in a recent sytematic review examining the effects of WBV on pulmonary function in COPD reported that no great benefits on pulmonary function were found similiarly to our results (1).
On the other hand there was only a significant change in FVC but not in FEV1 in the aerobic exercise group. In a recent meta anlaysis by Ora et al the effects of pulmonary rehabilitation including aerobic exercises in Noncystic Fibrosis Bronchiectasis, it was reported that The FEV1 assessment after PR between the active and control group did not show any significant increase and PR improves exercise tolerance in NCFB patients, but it has a modest impact on respiratory function (2). One study showed that limb cycle exercise at %75 of peak O2 resulted in significant improvement in FVC but not in any other spirometric parameters(3). This increase was explained as a result of improvement in inspiratory capacity and respiratory muscle strength. We also think that the improvement in FVC but not in FEV1 after aerobic exercise training can be result of an improvement in inspiratory capacity and increase in respiratory muscle strength as there might be an increase in maximal shortening of inspiratory muscles as an effect of training which might have led to an increase in inspiratory volume. But further studies with higher population focusing on spirometric parameters in bronchiectasis are needed to explain this.
References:
- Yang, X., Zhou, Y., Wang, P. et al. "Effects of whole body vibration on pulmonary function, functional exercise capacity and quality of life in people with chronic obstructive pulmonary disease: a systematic review." Clinical rehabilitation 30.5 (2016): 419-431.
- Ora J, Prendi E, Ritondo BL, Pata X, Spada F, Rogliani P. Pulmonary Rehabilitation in Noncystic Fibrosis Bronchiectasis. Respiration. 2022;101(1):97-105. doi: 10.1159/000517527. Epub 2021 Aug 5. PMID: 34352795
- Hsieh, Meng‐Jer,Lan CC, Chen NH et al. "Effects of high‐intensity exercise training in a pulmonary rehabilitation programme for patients with chronic obstructive pulmonary disease." Respirology 12.3 (2007): 381-388.
Question 14: How would you rationalize the significant decrease in mMRC score only in the WBV group?
Correction 14: Whole body vibration is known as dsypnea free exercise (1). According to the previous studies about the effects WBV in COPD, it was emphasized that due to lung emphysema and chronic bronchitis, COPD patients suffer from severe dyspnea especially during exercise. On the other hand, research has indicated that WBV exercise does not induce dyspnea in these patients (2). This may also be true for bronchiectasis patients.It was observed but not measured in the study that the aerobic exercise group especially at the first few treatment sessions dsypnea is induced with the exercise because of increasing oxygen demand. But in the WBV group perceived difficulty level was lower and the subjects in this group were adapted to exercise quickly. Some previous studies have reported that WBV exercise can,not just enhance physical status, but also manage the fatigue in various populations (3); which we think may also affect the dsypnea perception . The ‘‘tonic vibration reflex’’ caused by mechanical vibration induces a higher rate of motor unit recruitment in skeletal muscles by stimulating muscle spindles and Golgi cells. This provokes muscle contractions especially in the lower extremities and leads to an increase in muscle strength especially in lower extremities feeling less fatigue when compared with voluntary muscle contraction. In patients with stable COPD, it has been shown that WBV does not alter oxygen saturation (4). The oxygen saturation was measured at the begining and end of the exercise in both groups for monitoring the subjects during the execise but not anlysed as an outcome measure in this study. But we think another reason for lower dsypnea perception may be lower altretion in oxygen saturation in WBV group compared to aerobic exercise group.
References:
1- Furness, T, Joseph, C., Welsh, L et al. "Whole-body vibration as a mode of dyspnoea free physical activity: a community-based proof-of-concept trial." BMC research notes 6.1 (2013): 1-4.
2-Sañudo, B. Seixas, A., Gloeckl, R et al. "Potential application of whole body vibration exercise for improving the clinical conditions of covid-19 infected individuals: A narrative review from the world association of vibration exercise experts (wavex) panel." International Journal of Environmental Research and Public Health 17.10 (2020): 3650.
3-Alentorn-Geli, E.; Padilla, J.; Moras, G.; Haro, C.L.; Fernández-Solà, J. Six weeks of whole-body vibration exercise improves pain and fatigue in women with fibromyalgia. J. Altern. Complementary Med. 2008, 14, 975–981.
4-Furness, T.; Joseph, C.; Naughton, Get al.Benefits of whole-body vibration to people with copd: A community-based e_cacy trial. BMC Pulm. Med. 2014, 14, 38.
Question 15: Did you have sputum collected for the patients, any difference in microbiology? (for instance pseudomonas infection)
Correction 15: The sputum was collected by the pulmonologists for diagnosis and defining the severity of bronchiesctasis according to the severity index (When sputum cultures were examined, 10 patients in the WBV group had pseudomonas infection and 14 patients in the Aerobic group had pseudomonas infection ). But sputum collection was not an outcome measure fort he study.
Question 16: How do you think the fact that the exercise group were younger and had less exacerbations per baseline influenced your results?
Correction 16:Altgouh it is not statistically meaningful, the average ages of subjects in the groups were different.The older group was WBV but they tolarated the exercise well. The WBV is a commonly used exercise modality used in elderly for several reasons like significant improvement in dynamic and static balance, postural control, muscle strength, bone density and physical fitness (1-4).
If the average age of aerobic exercise group was older, the improvement in functional capacity might be lower as a result of age but the baseline funtional capacity is more important than age in terms of responses to exercise which is similar between two groups in our study (5).
References:
1-Cheung W, Mok, H. W., Qin, L et al. "High-frequency whole-body vibration improves balancing ability in elderly women." Archives of physical medicine and rehabilitation 88.7 (2007): 852-857.,
2- Rogan, S., Taeymans, J., Radlinger, L.et al"Effects of whole-body vibration on postural control in elderly: An update of a systematic review and meta-analysis." Archives of Gerontology and Geriatrics 73 (2017): 95-112
3-Bemben D, Stark, C., Taiar, R., et al. "Relevance of whole-body vibration exercises on muscle strength/power and bone of elderly individuals." Dose-Response 16.4 (2018): 1559325818813066.,
4-Gómez-Cabello, A., González-Agüero, A., Ara, I. et al. "Effects of a short-term whole body vibration intervention on physical fitness in elderly people." Maturitas 74.3 (2013): 276-278
5-Bouchard, Claude, and Tuomo Rankinen. "Individual differences in response to regular
physical activity." Medicine and science in sports and exercise 33.6; SUPP (2001): S446-S451
Question 17: Were patients asked to refrain from aerobic exercise at time of the study? Is it possible that some of the patients were also exercising in other times outside of the study?
Correction 17: The subjects in both groups did not attend any type of exercise program during the study.
Question 18: The discussion section needs to be re written, and English fixed.
Correction 18: It is revised.
Question 19: For the conclusion – Since there is profound evidence that aerobic exercise is beneficial in bronchiectasis, and by far less evidence for WBV, would you not be concerned that patients will prefer the "easy" intervention over the intervention that has clear advantages, not only for bronchiectasis but also for other cardiovascular aspects?
Correction 19: Infact, there is profound of evidence that pulmonary rehabilitation is beneficial in bronchiectasis. Athough aerobic exercise is one of the main components of pulmonary rehabilitation, the guidelines do not recommend just aerobic exercise for rhe rehabilitation of people with pulmonary diseases. The rehabilitation programs should include different types of exercise but aerobic exercise,strenthening and flexbility exercise are essential. Beside, after individual assessments, exercises for neuromotor control including balanace and coordination exercises and other types of exercise selected in accordance with the needs of patients can be added to rehabilitation programs. There is a growing evidence about the effects of WBV in various rehabilitation areas such as fibromyalgia, , elderly people, neurologic disorders, ortohpedic rehabilitation and pulmonary rehabilitaiton especially in COPD. The general indications for the use of WBVT like improving muscle power, balance, and bone density , are highly relevant in patients with pulmonary disease. The results of this study showed that WBV can be as affective as aerobic exercise also on dsypnea and functional capacity and can be suggested as an easy, shorter and enjoyable choice of exercise modality in the rehabilitation of bronchiectasis. The further studies with larger sample sizes are suggested to focus on the effects of WBV cardiopulmonary response,sptum clearance, fatigue, muscle strength, bone density and quality of life in patients with bronchiectasis.
Question 20: Extensive editing of English language and style required
Correction 20: Extensive editing of English language and style was edited.

Reviewer 2 Report
comment 1:
P values from Line 216-220 do not match with table1. kindly check and also place an asterisk against those significant p values, especially in Table 1.
comment 2:
expand certain abbreviations.
Author Response
Reviewers'comments:
Reviewer #2:
Dear Reviewer,
First of all thank you for your valuable contribution for improving the manuscript. We tried to revise the mansucript according to your suggestions.
Sincerely.
Comment 1: P values from Line 216-220 do not match with table1. kindly check and also place an asterisk against those significant p values, especially in Table 1.
Correction 1: Tables checked and corrected.
Comment 2: expand certain abbreviations.
Correction 2: We edited the abbreviations.

Round 2
Reviewer 1 Report
The writers related to all the points raised.
I continue to raise doubts about calling this study single- blinded. It is not clear to me what side you consider blinded. It is true that the patients do not know if they are considered controls or intervention group but they are definitely aware of what they are doing... as for the experimenters, any physiotherapist or technician performing spirometry, 6mwt or sit to stand test, can easily ask the patient if he was scheduled to do aerobic exercise or not.
However language and style remains the main problem of this manuscript:
Although English has been slightly improved, there are still many typing errors and significant style issues, for example, rationalizing WBV treatment (191-193) does not belong in the methods, or stating the inclusion, exclusion criteria as a list rather than a short paragraph or referencing to a box or figure with the criteria.
Another short example - in line 299 "it was shown that 6mwt and ftsst were significant in both groups" I assume that this means that 6mwt and fttst were significantly improved at the end of the study compared to the results prior to the study...
Another example - (lines 415-417)
"Considering our study, the age difference between the two groups was not significant, but the age difference was large. This may be related to the fact that the quality of life of elderly patients is more easily affected, and recovery is more difficult"
I am not sure how the second sentence relates to the first, and what the meaning of a large difference that is not significant means...
A conclusion is usually a short paragraph describing what can be learned from the study rather than discussing evidence from other literature.
Overall, the discussion is very long and cumbersome making it very difficult for the reader to follow.
I would therefore suggest thorough editing of the manuscript before publishing
Author Response
Dear Reviewer,
First of all, thank you for your valuable contribution to improving the manuscript. We tried to revise the manuscript according to your suggestions and for some of them, we tried to write the possible explanations as listed below.
Sincerely.
1: I continue to raise doubts about calling this study single-blinded. It is not clear to me what side you consider blinded. It is true that the patients do not know if they are considered controls or intervention group but they are definitely aware of what they are doing... as for the experimenters, any physiotherapist or technician performing spirometry, 6mwt, or sit to stand test, can easily ask the patient if he was scheduled to do aerobic exercise or not.
- We mean, the assessor physiotherapist knowing the methodology of the study, who was experienced in the field of pulmonary rehabilitation for about 20 years as a clinician, researcher having studied in PR and academic staff, was blinded for the groups of subjects and we ensure that she did not ask the patients in which group they were as it should be… Also, the patients do not know if they are considered controls or intervention groups. But if it may raise doubts about calling the study single-blinded for you and also readers we can present the study as a randomized controlled study (The revision about this has been made in the resubmitted manuscript).
2: However language and style remain the main problem of this manuscript:
Although English has been slightly improved, there are still many typing errors and significant style issues, for example, rationalizing WBV treatment (191-193) does not belong in the methods, or stating the inclusion, and exclusion criteria as a list rather than a short paragraph or referencing to a box or figure with the criteria.
Another short example - in line 299 "it was shown that 6mwt and FTSTT were significant in both groups" I assume that this means that 6mwt and fttst were significantly improved at the end of the study compared to the results prior to the study.
Another example - (lines 415-417)
"Considering our study, the age difference between the two groups was not significant, but the age difference was large. This may be related to the fact that the quality of life of elderly patients is more easily affected, and recovery is more difficult"
I am not sure how the second sentence relates to the first, and what the meaning of a large difference that is not significant means...
-The language and the style of the manuscript have been revised.
WBV treatment has been explained under the sub-title of Pulmonary Rehabilitation
The inclusion and exclusion criteria have been stated in a short paragraph The sentence in line 299 has been changed to “When the results before and after the treatment were compared, it was shown that there were significant increases in terms of 6MWT and FTSST in both groups”.
For lines 415-417 the sentences have been revised as “Also improvements in QoL were achieved in the WBV group when comparing before and after treatment, although it was not significant. One disadvantage of questionnaires is that they are based on patients’ statements which can be related to the patient's level of satisfaction and what the patient expects from the treatment. This can be a reason for the difference in QoL between the two groups.
3: A conclusion is usually a short paragraph describing what can be learned from the study rather than discussing evidence from other literature.
-The conclusion has been shortened as
The results of this study showed that WBV can be as effective as aerobic exercise on dyspnea and functional capacity, and is suggested as an easy, shorter, and enjoyable choice of exercise modality in the rehabilitation of bronchiectasis. Larger studies are needed to define the optimal intensity and duration of WBV, as well as to investigate its possible long-term effects
4: Overall, the discussion is very long and cumbersome making it very difficult for the reader to follow.
- The discussion has been shortened and revised. We revised the references.
